# Reducing blood product wastage: Insights from a retrospective study in a tertiary health facility

Jawaher Alsughayyir[1]*, Afnan Alghamdi[2], Abdulaziz M. Almuqrin[1], Abdulrahman Alshalani[1], Nasser Alqahtani[3], Ammar Alsughayir[4], Alyazeed Alsaif[4], Yazeed Alfalah[5], Sarah Bakr Abobakr[4], Naif Bin Muhannaa[4], Fahad Alshehri[6,7]

1 Department of Clinical Laboratory Sciences, College of Applied Medical Sciences, King Saud University, Riyadh, Saudi Arabia, 2 Regional Laboratory and Blood Bank, Al-Baha Health Cluster, Ministry of Health, Saudi Arabia, 3 Najran University, Medicine College, Family Medicine Department, Najran, Saudi Arabia, 4 King Fahad Medical City, Clinical Pathology and Laboratory Medicine, Transfusion Medicine Department, Saudi Arabia, 5 Qassim University Medical City, Laboratory and Blood Bank Department, Blood Bank, Buraidah, Saudi Arabia, 6 King Faisal Medical City for Southern Regions, Pathology and Clinical Laboratory Medicine Administration, Hematology Department, Abha, Saudi Arabia, 7 Aseer Health Cluster, Aseer Central Hospital, Laboratory and Blood Bank Department, Hematology Section, Abha, Saudi Arabia

* jalsughayyir@ksu.edu.sa

## Abstract

### Introduction

Due to the absence of prior institutional assessments and limited local studies lacking key performance indicators, this study examines the operational efficiency of the blood bank at King Fahad Medical City (KFMC) in Riyadh, Saudi Arabia, and aims to identify inefficiencies in blood ordering practices and recommend corrective measures.

### Methods

This retrospective study analyzed KFMC blood bank inventory data from January 2022 to December 2023, using descriptive statistics and three key performance indicators: the crossmatch-to-transfusion ratio (C/T), wastage as a percentage of issue (WAPI), and the issuable stock index (ISI).

### Results

Overall, 95.05% of blood products were utilized, 3.8% were returned in acceptable condition, 4.2% expired, and 0.6% were discarded for non-expiry-related reasons. The C/T ratio was 1.5, the ISI was maintained at 1.1 across all blood groups, and WAPI remained below 1.2%, indicating efficient inventory management.

### Discussion

Blood bank management at KFMC demonstrated strong operational efficiency through strategic inventory management. Most post-crossmatch wastage resulted

**Data availability statement:** All relevant data are within the paper and its Supporting Information files.

**Funding:** This work was supported by King Fahad Medical City (KFMC), Riyadh Second Health Cluster, (Grant No. IRF 024-051). The funder had no influence or role in the study's design, data analysis, interpretation, writing, or publication decisions.

**Competing interests:** The authors have declared that no competing interests exist.

from over-ordering in high-demand areas such as the intensive care unit, reflecting the urgency of these settings. Regular audits and updates to the Maximum Surgical Blood Ordering Schedule are recommended to optimize efficiency and minimize waste. This study provides a valuable benchmark for future research on blood bank operations and inventory management.

## Introduction

The inherent challenges of balancing supply and demand, coupled with the limited shelf life of certain blood products, have contributed to substantial disposal within blood banks. Therefore, developing an effective inventory management framework is critical for optimizing the blood supply chain. Inventory inefficiencies may arise from expiration, inadequate stock turnover, suboptimal storage or handling, and excessive ordering practices, all of which can increase wastage and operational burden. External factors such as pandemics, natural disasters, and fluctuations in donor availability may further disrupt supply–demand equilibrium and compromise product availability [1–5]. The implementation of a comprehensive set of operational key performance indicators (KPIs) is critical to ensure that blood bank functions adhere to established performance standards across multiple domains, including quality management, blood component processing, and transfusion services [6]. In addition to KPIs, number of laboratory indices and parameters have also been proposed by the Blood Stocks Management Scheme (BSMS) to monitor and improve the performance of blood bank management [7]. For instance, the crossmatch-to-transfusion (C/T) ratio serves as an indicator of blood utilization [8]. Other indices include the issuable stock index (ISI) which estimates the daily stock based on current usage [9], the wastage as a percentage of issue (WAPI) to determine wastage rates of blood products [9], the transfusion index (Ti) which represents the adequacy of the number of units subjected to crossmatching [10], and the transfusion probability (TP%) which reflects blood utility [11], among other parameters with each reflecting a specific aspect of blood bank inventory management.

Few studies have examined inventory management, utilization, or wastage in Saudi healthcare. Existing research typically assesses single variables within individual facilities or focuses on blood supply-demand dynamics [ 10,12–14]. Nationwide analyses are limited to MOH-affiliated central and peripheral blood banks, excluding major medical cities such as those in Riyadh's second health cluster [10,15,16]. King Fahad Medical City (KFMC), is a 1,200-bed tertiary care facility integrated within Riyadh's Second Health Cluster.. KFMC's blood bank is responsible for supplying blood to various departments and units, including the emergency department, internal wards, surgery wards, intensive care units, outpatient clinics, gynecology and obstetrics wards, chemotherapy and cancer wards. The aim of this study is to investigate the current management of blood bank operations at KFMC by assessing the overall blood utilization practices and inefficiencies. Findings of this study would

hold potential utility in optimizing blood transfusion services and facilitating the implementation of strategic measures to improve the management and efficiency of blood bank operations.

## Materials and methods

### Study design

This retrospective study was conducted using data retrieved from the blood bank database at KFMC in Riyadh, Saudi Arabia, comprising records from January 2022 to December 2023. These were the most recent years with accessible records. Data from earlier years were archived using a different database system, making information retrieval challenging.

The collected data comprised records of the collection date of each blood component, the number of units issued, the original department responsible for the request, the number of units utilized, expired, discarded, and the reason for discard for each blood product. All data were fully anonymized, and at no stage during or after the data collection process did any of the authors have access to personally identifiable information. Transfusion-transmitted infection reactive blood units were excluded from the study. Data were gathered from ninety-eight departments and wards and organized into nine broader departments to enhance the clarity of the reported analyses (S1 Table).

Within the scope of this study, discarded blood units are defined as blood product units collected but not used for any reason (e.g., clot formation, leakage, without cord or enough segments, etc.), blood wastage was defined as blood products that were cross-matched but not transfused for any reason (e.g., delay in time, improper handling, etc.) and subsequently not reassigned to an alternative patient or recipient. Expired units were defined as those reaching the expiry date without being used. Returned blood products are handled according to institutional policies that require verification of temperature integrity, inspection for physical abnormalities, and documentation of chain-of-custody before determining suitability for reissue. The study protocol was approved by the Institutional Review Board of KFMC before its commencement (IRB 24–254).

### Statistics

This descriptive study displayed the number of blood product units collected, issued, expired, discarded, or returned as frequencies or percentages. Maximum surgical blood ordering schedule (MSBOS) was calculated according to Mead's criterion [17] and is approved and regulated by the hospital's transfusion committee.

To evaluate departmental differences in blood product utilization and returns, the Kruskal–Wallis test was conducted, as the data did not meet the assumptions of normality. Post hoc analysis (e.g., Dunn's test with Bonferroni correction) was performed to identify specific interdepartmental differences. *P*-value less than 0.05 was considered statistically significant. All statistical analyses were performed by GraphPad Prism v9.2.0 (GraphPad Software, Inc., San Diego, CA, USA). Three blood bank indices were calculated in this study: C/T ratio as an indicator of blood bank utilization according to the following formula:

$$\frac{C}{T} = \frac{Number\ of\ units\ crossmatched}{Number\ of\ units\ transfused} \tag{1}$$

C/T ratio less than 2 is considered optimal, whereas a ratio exceeding 2 suggests over-ordering of blood products according to the guidelines established by the Association for the Advancement of Blood and Biotherapies (AABB) [18]. The second index was WAPI [9]. Although no universal threshold exists, previous studies considered values 2–7% to reflect optimal inventory management [19–21], with higher values indicating greater levels of wastage.

$$WAPI = \frac{Number\ of\ units\ wasted}{Number\ of\ units\ issued} \times 100 \tag{2}$$

ISI was calculated to estimate the average daily inventory of available blood units according to the following formula [22]:

$$ISI = \frac{Issuable\ stock\ (number\ of\ unreserved\ RBC\ units\ of\ all\ groups)}{Nominal\ stock\ (number\ of\ issued\ RBCs\ during\ the\ \frac{year}{365})} \qquad (3)$$

Maintaining approximately five days of issuable stock is generally advised [21].

## Results

### Inventory management and blood utilization trends

A distinct annual utilization pattern was observed for each blood component across the departments, with trends remaining consistent for each department over the study period. Since the patterns of utilization and wastage of all blood products were similar across the two years (Table 1S), the data from both years were combined and reported as a cumulative total (Table 1).

At KFMC, the number of blood units issued varied depending on the requesting department, where requests are typically fulfilled based on the ordering ward's needs. For surgical procedures, requests are determined on a case-by-case basis according to the MSBOS, though exceptions are made for massive transfusion protocols and emergency situations, allowing for additional units beyond standard limits. A total of 40,086 units of packed red blood cells (pRBCs) were prepared and available in the inventory, of which 39,230 units (97.8%) were issued, averaging as 19,615 pRBC units per year. Over the same period, a total of 16,028 units of FFP were collected, with 15,008 units (93.6%) being issued, averaging as 7,504 FFP units per year in a 1,200-bed capacity hospital. Utilization patterns for pRBCs and FFP were generally similar, with the main hospital wards representing the highest request levels, accounting for 47.25% of pRBCs and 61.4% of FFP consumption (Table 2).

Further analysis of hospital main wards utilization revealed that the main hospital intensive care unit (MH ICU) had the highest use of both pRBCs 13.4%, (5,389 units) and FFP (31.8%, 5,098 units). Additionally, the Women's specialized hospital (WSH) demonstrated high demand for pRBCs and FFP, accounting for 15.25% (6,121 units) of total pRBCs usage and 12.7% (2,040 units) of FFP. The emergency and accident department also demonstrated high utilization, with 10.9% (4,380 units) of pRBCs and 6.5% (1,05 units) of FFP (Table 2). For platelets, 18,133 units were registered in the inventory, of which 16,366 units were issued, reflecting an overall utilization rate of 90.25% (Table 1). The primary consumers of platelets were the main hospital wards, which collectively utilized 52.7% (9,566) of all issued platelet units, and

**Table 1. Total number (N) and frequencies (%) of utilized blood products per department.**

| | | Adult CVOR* | AED | CSH | OR | Main hospital wards | Null | Pediatric departments | Rehab Hospital wards | WSH wards | Total | *P* |
|---|---|---|---|---|---|---|---|---|---|---|---|---|
| **RBCs** | N | 112 | 4,380 | 4,408 | 375 | 18,949 | 3,323 | 1,559 | 859 | 6,121 | 40,086 | 0.0006 |
| | % | 0.2 | 10.9 | 10.9 | 0.9 | 47.25 | 8.2 | 3.8 | 2.1 | 15.25 | 100 | |
| **FFP** | N | 144 | 1,052 | 1,076 | 206 | 9,853 | 1,110 | 283 | 264 | 2,040 | 16,028 | 0.0051 |
| | % | 0.9 | 6.5 | 6.7 | 1.2 | 61.47 | 6.9 | 1.7 | 1.6 | 12.7 | 100 | |
| **Platelets** | N | 56 | 636 | 3,463 | 37 | 9,566 | 1,920 | 647 | 327 | 1,481 | 18,133 | 0.0005 |
| | % | 0.3 | 3.5 | 19 | 0.2 | 52.7 | 10.5 | 3.5 | 1.8 | 8.1 | 100 | |
| **Cryoprecipitate** | N | 804 | 353 | 652 | 57 | 3,303 | 509 | 138 | 42 | 281 | 6,139 | 0.008 |
| | % | 13.1 | 5.75 | 10.6 | 0.9 | 53.8 | 8.2 | 2.25 | 0.6 | 4.5 | 100 | |

Analysis of departmental utilization patterns was determined using the Kruskal–Wallis test (P < 0.05 is considered statistically significant).

* CVOR: cardiovascular operating room; AED: accidents and emergency department; CSH: children's specialized hospital; OR: main operating room; Null: untraceable; WSH: women specialist hospital wards.

**Table 2. Total number (N) and frequencies (%) of blood products collected, issued, expired, wasted, and returned during the study period (January 2022 – December 2023).**

| | RBCs | | FFP | | Platelets | | Cryoprecipitate | | Total | |
|---|---|---|---|---|---|---|---|---|---|---|
| | N | % | N | % | N | % | N | % | N | % |
| Collected | 40,086 | 100 | 16,028 | 100 | 18,133 | 100 | 6,139 | 100 | 80,386 | 100 |
| Issued | 39,230 | 97.8 | 15,008 | 93.6 | 16,366 | 90.2 | 5,813 | 94.6 | 76,417 | 95 |
| Expired | 609 | 1.5 | 861 | 5.3 | 1,704 | 9.4 | 252 | 4.1 | 3,426 | 4.2 |
| Wasted* | 247 | 0.6 | 159 | 0.9 | 63 | 0.3 | 69 | 1.1 | 538 | 0.6 |
| Returned† | 2,550 | 6.3 | 320 | 2 | 123 | 0.6 | 126 | 2.05 | 3,119 | 3.8 |

\* Issued but not used and not returned.

† Returned to the blood bank after issue in acceptable condition.

the children's specialized hospital (CSH), accounting for 3,463 units (19.0%). A total of 6,139 units of cryoprecipitate were available in the inventory, with 5,813 units issued, representing 94.6% of the inventory. Utilization was highest in the main hospital wards, which accounted for 53.8% (3,033 units), followed by the MH ICU with 1,651 units (26.8%) (S1 Table). These findings underscore the essential role of cryoprecipitate in cardiovascular surgical procedures.

The utilization of all four blood components varied significantly across the nine hospital departments, highlighting distinct patterns in blood product consumption. (Table 2). Analysis of interdepartmental patterns for each blood product revealed that only FFP showed statistically significant differences in utilization between the main hospital wards and both the emergency and rehabilitation departments ($P = 0.032$, $P = 0.035$, respectively). While other blood products exhibited significant overall departmental differences in usage, these were not statistically significant, likely due to variability in the size and distribution of sub-departments within the nine main hospital divisions (Table 2).

Platelets demonstrated the highest rate of expiry, accounting for 9.4% of the inventory (1,704 units), followed by FFP (861 units, 5.3%), cryoprecipitate (252 units, 4.1%), and pRBCs (609 units, 1.5%) (Table 1). These findings underscore the predominance of pRBC in utilization and the vulnerability of platelets to disposal due to their short half-life. Although the expiry rate of platelets was relatively low, these findings highlight the need for strategies to minimize the disposal of frequently used components.

The overall C/T ratio for the period 2022–2023 was 1.5, indicating efficient blood utilization and effective transfusion management. All four blood products demonstrated efficient storage durations before dispatch. PRBCs had an average storage duration of $14.94 \pm 8.9$ days from the time of collection. The average corresponds with the estimated ISI for 2022 and 2023, which quantify the daily availability of RBC units for each blood group (Table 3). FFP had an average storage

**Table 3. Issuable Stock Index (ISI) for each blood group during the study period (January 2022 – December 2023.**

| Blood Group | 2022 | | | 2023 | | |
|---|---|---|---|---|---|---|
| | Issuable stock | Nominal stock | ISI | Issuable stock | Nominal stock | ISI |
| O Positive | 23.75 | 21.5 | 1.10 | 27.7 | 25.1 | 1.10 |
| O Negative | 2.57 | 2.34 | 1.01 | 3.07 | 2.79 | 1.10 |
| A Positive | 10.84 | 9.84 | 1.10 | 15.2 | 13.8 | 1.10 |
| A Negative | 0.97 | 0.88 | 1.10 | 1.67 | 1.51 | 1.10 |
| B Positive | 9.08 | 8.24 | 1.10 | 9.63 | 8.75 | 1.10 |
| B Negative | 0.60 | 0.55 | 1.09 | 0.84 | 0.76 | 1.10 |
| AB Positive | 1.65 | 1.50 | 1.1 | 1.98 | 1.80 | 1.10 |
| AB Negative | 0.08 | 0.07 | 1.14 | 0.07 | 0.06 | 1.16 |

duration of 21.4±27.21 days, while platelets, due to their inherently shorter lifespan and higher susceptibility to aggregation, had a shorter average of 2.96±1.1 days, and cryoprecipitate had an average of 50.97±39.6 days.

## Causes of Blood Discard

Figure 1 summarizes the primary reasons for blood component discards. Discarded units refer to blood components rejected after collection and before crossmatch due to collection-related issues. Of the 40,086 collected pRBC units, 259 units (0.6%) were discarded, primarily due to hematology or serology borderline test results (23.9%) and elapsed time from extended room temperature exposure (22.7%). The discard rate for FFP was 0.9% (159 units) of the inventory, with insufficient storage space accounting for the majority of losses (36.4%). Platelet discards totaled 63 units (0.3%), predominantly due to low quantities (<10% of the desired volume). Cryoprecipitate discards amounted to 69 units (1.1%), mainly due to insufficient storage capacity (Fig 1). WAPI values for pRBCs, FFP, and platelets were 0.6%, 1.05%, and 0.3%, respectively. Cryoprecipitate showed the highest WAPI among all blood components at 1.1%.

## Departmental wastage disparities

KFMC has an effective policy allowing departments to return unused blood products in acceptable condition, facilitating their future utilization and significantly reducing disposal. Most departments effectively utilized the allocated blood components as requested, with only a combined total of 3,119 units being returned after issue, which is equivalent to 4% of all issued blood products (Table 1). A total of 2,550 units of packed red blood cells (pRBCs) were returned, representing the majority of all returned blood components. Statistical analysis revealed a significant difference in the number of returned pRBC units among departments (*P*=0.0005; Fig 2). Although the main hospital wards accounted for the highest proportion of pRBC returns (52.7%), the only statistically significant pairwise difference was identified between the rehabilitation

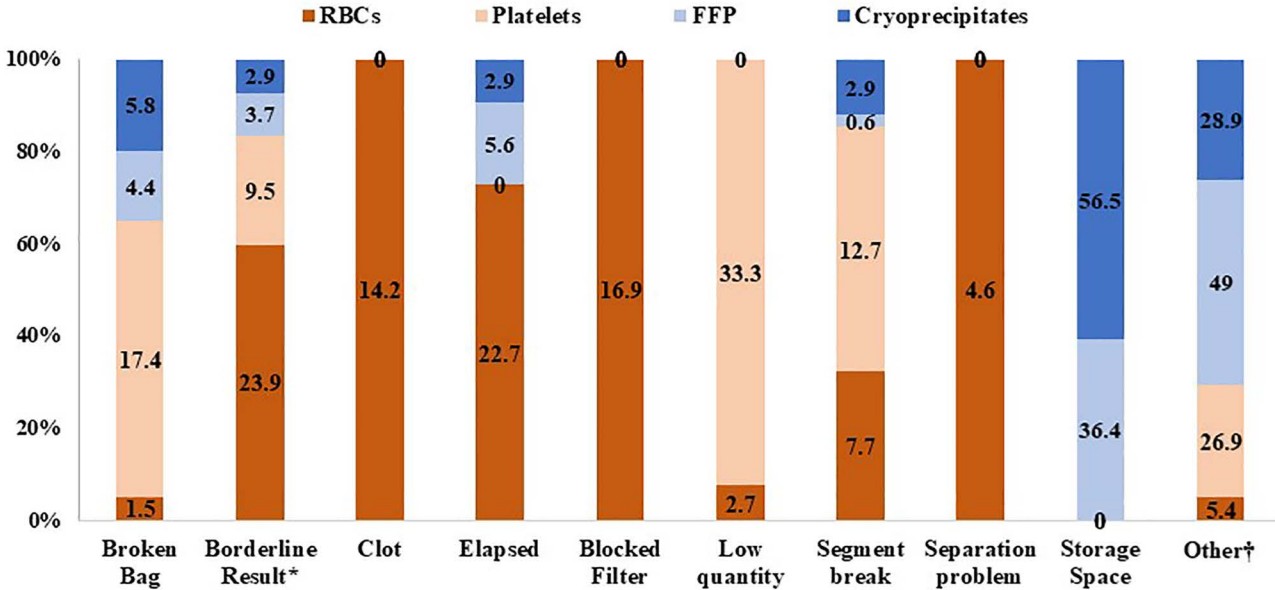

**Fig 1. Stacked column chart illustrating the proportional distribution of blood product wastage reasons across different blood components.** Each column represents a wastage reason, with colored segments indicating the percentage contribution of individual blood products. Percentages are normalized within each wastage reason. *Any hematology or serology value that is near the threshold for a positive result. †ABO blood grouping issues, air in the bag, quality control reasons, fat plasma, or red cell volume exceeds standard limits.

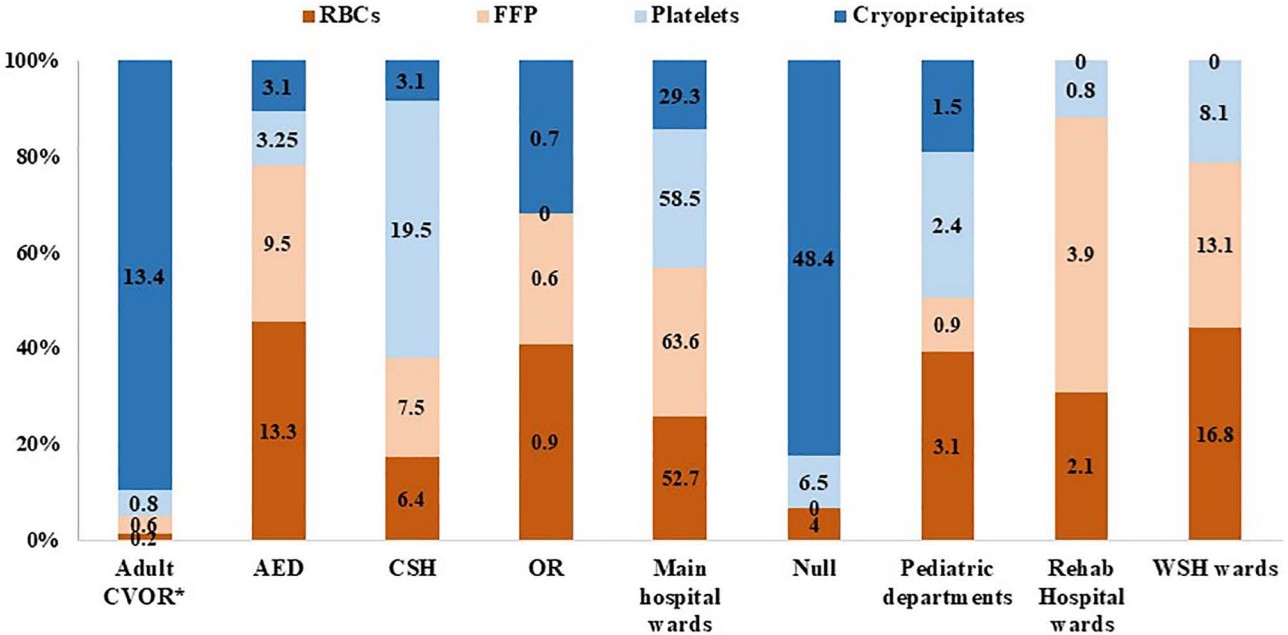

**Fig 2. Percentage-stacked column chart showing the distribution of returned blood products across hospital departments.** Each column displays the normalized percentage of returned blood products. ˚CVOR: cardiovascular operating room; AED: accidents and emergency department; CSH: children's specialized hospital; OR: main operating room; Null: untraceable; WSH: women specialist hospital wards.

and accident and emergency departments ($P = 0.041$). No other interdepartmental comparisons demonstrated significant differences.

Only 320 units of FFP were returned, with 63.6% of these units originating from the main hospital wards, and the ICU sub-department contributing 24.3% of this total. Statistical analysis revealed a significant difference in the number of returned FFP units among departments ($P = 0.004$), possibly influenced by the statistically significant pairwise difference identified between the children's hospital departments and the women's health department ($P = 0.0235$). The number of returned units of platelets and cryoprecipitate was lower, with a total of 123 units of platelets and 126 units of cryoprecipitate. Most platelets returned originated from the main hospital wards (58.5% of all returned platelets), and a detailed analysis revealed that most returns were particularly from the ICU sub-department (13.8%). Similarly, 29.3% of cryoprecipitate returns were from the main hospital wards. These figures reflect the critical and time-sensitive nature of healthcare services provided in these departments, accounting for the relatively higher volumes of returned units. These are reasonable figures considering the critical and time-sensitive demands of the requesting departments.

## Discussion

Monitoring blood product disposal against local and national benchmarks is critical to maintaining optimal inventory levels. This study is the first to examine the ISI alongside WAPI and C/T ratio as comprehensive indicators of blood bank performance in Saudi Arabia. These findings lay the groundwork for future research aimed at establishing benchmarks and sharing effective strategies to optimize inventory management and minimize excessive disposal in transfusion services.

Our analysis demonstrates that the inventory management system of the blood bank at KFMC was executed with a high degree of efficiency. In terms of utilization, 95% of all blood components were allocated, with only 3.8% of issued units returned to the blood bank in good condition. For pRBCs, reissue was permitted only when temperature integrity

was verified using indicator strips and container seals remained intact. Returned units were not included in the utilization or disposal analyses, as their subsequent reallocation or disposal could not be reliably tracked through the IT inventory system. Collectively, these findings reflect a deliberate inventory strategy that maintains a reserve stock equivalent to 10% of the average issued units for each blood product, ensuring readiness for emergencies or crises.

ISI levels were lower than those typically recommended in healthcare institutions of comparable capacity [21]. For instance, In England, Wales, and Northern Ireland the ISI levels can range from 6.6 to 9.2 depending on the hospitals crossmatch reservation period [22,23]. Nevertheless, maintaining the lower ISI levels observed in this study have improved inventory control. Consistent with this, Park et al. demonstrated that replacing the uniform 5-day ISI used across Korean hospitals with a target ISI of 1 day could improve blood inventory management and reduce blood component wastage [24].

The institution adopted the MSBOS, a standardized guideline outlining the typical number of blood product units required for various elective surgical procedures [25–27]. This approach has significantly optimized blood management practices and is regularly approved by the Transfusion Review Committee at the institute, ensuring adherence to best practices in transfusion medicine. In addition, the enforcement of a 'first-in, first-out' policy minimized blood unit disposal due to expiration. The practice of returning unused blood products for reallocation significantly contributed to efficient resource utilization. To further enhance inventory management, the institution disposes of excess FFP of blood group O+ when demand for it is low. This practice helps maintain adequate storage space for fresher units, ensuring readiness to meet ongoing requirements. By restricting early ordering, this approach helps preserve the quality of blood products by minimizing prolonged exposure to room temperature.

The WAPI for all blood products remained ≤1.2% indicating efficient blood bank management and alignment with international benchmark targets [28]. The reported WAPI levels compare favorably with international benchmarks, where median pRBC values range from 1.3% to 2.4% across Europe, North America, and other regions [29]. It is also substantially lower than rates reported in a tertiary healthcare center in Jeddah, where WAPI values for pRBCs, FFP, and platelets reached 12–9%, 22–8%, and 35–20%, respectively, underscoring the efficiency of KFMC's inventory management [30]. Notably, certain regions in Saudi Arabia have reported wastage rates as high as 89.2% [31]; however, interpreting this variability is challenging due to differences in locally implemented MSBOS strategies, donation rates, inventory constraints, and other logistical complexities, further compounded by the absence of a centralized national blood bank management system for standardization and comparison. Consistent with our findings, MSBS guidelines recommend WAPI targets of <2.5–7 for red blood cells and <4–7 for platelets, with no defined thresholds for plasma or cryoprecipitate due to longer storage stability [21]. The conservative wastage rates at KFMC can be attributed to the effective storage conditions and handling protocols. The hospital utilizes cooling containers with portable thermometers to ensure optimal storage conditions for blood products during transit, preventing exposure to ambient temperatures and maintaining product integrity which otherwise can affect the quality of blood products [32,33].

The reported C/T ratio of 1.5 is well within the widely accepted benchmark of ≤2 [18]. This value is comparable to C/T ratios reported from tertiary centres in Saudi Arabia (1.8) [34] and rural tertiary hospitals in India (1.26) [35], and notably lower than that observed in Sudan (4.4) [35]. No local studies have been conducted that specifically investigated WAPI alongside C/T; thereby limiting the ability to compare these parameters with other regional benchmarks. This positions the current study as a valuable benchmark for future research on blood bank inventory systems.

The tendency of over-ordering was mostly noted among patients at the AED and OR departments. The preoperative requisition of the blood units is mostly based on the assumption of a worst-case scenario or overestimation of intra-operative or post-operative blood loss, which leads to unnecessary demands for a large number of pRBCs [36]. The overall disposal rate due to technical or handling issues was low (0.6%), indicating effective management. This rate is comparable to previously reported single-center reports [37], although higher rates have been documented in other tertiary healthcare hospitals in Saudi Arabia which reached 19.44% [30]. In line with existing literature, platelets were the

most frequently discarded component with rates ranging from approximately 18% to 38%, reflecting their short shelf-life [30,37,38]. Notably, borderline serological results were the predominant cause of disposal, underscoring stringent screening practices. In contrast, disposals due to time elapsed (16.6%) and limited storage capacity highlight areas for further optimization. Understanding these department-specific dynamics enables targeted interventions, such as refining MSBOS and enhancing communication, to reduce inefficiencies and enhance system-wide performance [4,25]. Predictive tools, including machine-learning models using real-time clinical data, further improve transfusion forecasting and reduce unnecessary reservations [39,40].

Although platelets are often associated with high disposal rates due to their inherently short shelf-life, their efficient utilization reflected exemplary operational practices. They still recorded the highest expiration rate (9.4% of inventory), but effective management, such as returning units in acceptable condition and minimizing discards due to mishandling, reflects the success of the measures used in handling this fragile blood product. On an international level, studies examining blood product discard rates in 17 European countries reported that the average platelets disposal rate was 13% and varied from 6.7% to 25% across institutions [41–44]. KFMC currently upholds effective inventory management; however, a number of recommendations can enhance the efficiency of blood bank management. For instance, engaging staff to identify causes of collection-related issues, such as piercing errors, may further mitigate unnecessary disposal [45]. In addition, effective communication with clinical staff plays a crucial role in enhancing ordering practices. Even modest reductions in wastage can produce substantial cost savings, particularly in high-volume centers [30]. Beyond economic considerations, excessive disposal may indirectly affect patient care by constraining inventory during periods of increased demand. In addition, skilled and well-trained transfusion laboratory staff, electronic crossmatching, inventory transparency, and streamlined management procedures were crucial for blood bank management [28].

This study is limited by its retrospective, single-center design and relatively short study period, which may constrain generalizability; additionally, reliance on existing records may result in incomplete or inconsistent information. Given the scarcity of local research from hospitals of comparable capacity, these findings nonetheless provide an important reference point for future investigations into blood bank operations and inventory management.

## Conclusion

This article highlights the management strategies of KFMC's blood bank, focusing on its effectiveness in balancing utility with expiration dates to minimize the wastage rate. These findings provide practical insights that can guide comparable institutions in optimizing inventory management, refining ordering practices, and reducing wastage through evidence-based, department-specific interventions. Storage-related constraints highlight the need for future work on enhancing storage capacity and inventory infrastructure to support more efficient blood management.

## Supporting information

**S1 Table. Summary of data collected from ninety-eight departments aggregated into nine broader categories.** (DOCX)

## Author contributions

**Conceptualization:** Jawaher Alsughayyir, Abdulaziz M. Almuqrin, Abdulrahman Alshalani, Fahad Alshehri.

**Data curation:** Alyazeed Alsaif, Yazeed Alfalah, Sarah Bakr Abobakr.

**Formal analysis:** Jawaher Alsughayyir, Nasser Alqahtani, Alyazeed Alsaif, Sarah Bakr Abobakr.

**Funding acquisition:** Naif Bin Muhannaa, Fahad Alshehri.

**Methodology:** Abdulaziz M. Almuqrin, Yazeed Alfalah.

**Project administration:** Naif Bin Muhannaa.

**Supervision:** Ammar Alsughayir.

**Visualization:** Nasser Alqahtani, Ammar Alsughayir.

**Writing – original draft:** Jawaher Alsughayyir, Afnan Alghamdi.

**Writing – review & editing:** Jawaher Alsughayyir, Afnan Alghamdi, Abdulrahman Alshalani, Fahad Alshehri.

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
