## [Decision Letter · Decision Letter 0]

8 Dec 2025

Dear Dr. Alsughayyir,

Thank you for submitting your manuscript to PLOS ONE. After careful consideration, we feel that it has merit but does not fully meet PLOS ONE’s publication criteria as it currently stands. Therefore, we invite you to submit a revised version of the manuscript that addresses the points raised during the review process.

We look forward to receiving your revised manuscript.

Kind regards,

Mohammad Shokouhifar

Academic Editor

PLOS One

Journal Requirements:

“This work was supported by King Fahad Medical City (KFMC), Riyadh Second Health Cluster, (Grant No. IRF 024-051).”

5. Please include a copy of Table 4 and 5 which you refer to in your text on page 10.

Reviewer's Responses to Questions

**Comments to the Author**

1. Is the manuscript technically sound, and do the data support the conclusions?

Reviewer #1: Yes

Reviewer #2: Partly

Reviewer #3: Yes

2. Has the statistical analysis been performed appropriately and rigorously?

Reviewer #1: Yes

Reviewer #2: Yes

Reviewer #3: Yes

3. Have the authors made all data underlying the findings in their manuscript fully available?

Reviewer #1: Yes

Reviewer #2: No

Reviewer #3: Yes

4. Is the manuscript presented in an intelligible fashion and written in standard English?

Reviewer #1: Yes

Reviewer #2: Yes

Reviewer #3: Yes

Reviewer #1: The manuscript titled “Reducing Blood Product Wastage: Insights from a Retrospective Study in a Tertiary Health Facility” investigates the operational efficiency of the blood bank at King Fahad Medical City (KFMC) in Riyadh, Saudi Arabia, with a focus on identifying inefficiencies in blood ordering and minimizing wastage. The study uses retrospective data from January 2022 to December 2023 to assess blood product utilization and wastage, employing key performance indicators such as the crossmatch-to-transfusion ratio (C/T), wastage as a percentage of issue (WAPI), and the issuable stock index (ISI). The results indicate efficient blood product utilization, with wastage rates below international benchmarks, and the paper provides recommendations for improving inventory management practices at KFMC. Authors need to address the following comments:

1. The introduction could be more concise, especially the sections that describe previous studies, to avoid redundancy with later discussions in the manuscript.

2. The background does not explicitly define the threshold values for C/T, ISI, and WAPI, which are crucial for understanding the operational standards and benchmarks used in the analysis.

3. There is no discussion on the limitations of using retrospective data; a brief acknowledgment of this limitation would strengthen the methodology section.

4. The manuscript does not clarify why data from January 2022 to December 2023 was selected as the study period, given the changes in blood demand and hospital practices over time.

5. The study claims the C/T ratio is 1.5, but a brief explanation of how this compares to international standards or previous studies would help contextualize the result.

6. The methods section briefly mentions the Kruskal–Wallis test for comparing departmental utilization, but does not justify why this non-parametric test was chosen over other options, such as ANOVA.

7. While the paper discusses the concept of WAPI, it would be helpful to compare this metric with benchmarks from similar hospitals to evaluate the true significance of the reported values.

8. The significance of the 0.6% discard rate is not well contextualized; the manuscript could benefit from a comparison to other similar healthcare facilities or regions.

9. The relationship between the blood bank’s operational efficiency and the specific departments (ICU, surgery, etc.) could be explored in more detail to provide deeper insights into the causes of inefficiencies.

10. The discussion of over-ordering, particularly in high-demand departments like the ICU, would benefit from further elaboration on possible solutions, such as implementing predictive algorithms for blood demand forecasting.

11. The paper would benefit from including more specific recommendations for improving the MSBOS, particularly in departments with high over-ordering rates.

12. The methodology mentions the collection of anonymized data, but no information is provided about how the authors ensured the quality and consistency of the data across different departments.

13. While the study mentions that certain blood components were returned in acceptable condition, it would be useful to discuss whether these returned products were reallocated or disposed of.

14. A potential limitation is the focus on a single hospital; the authors should address the generalizability of their findings to other hospitals in Saudi Arabia or internationally.

15. The manuscript provides data on blood product wastage but does not discuss potential consequences of waste, such as increased costs or impacts on patient care.

16. The paper mentions that “not enough storage space” contributed to some discard rates; this could be a key area for future research on optimizing storage management.

17. There is little information on the process followed for managing returned blood products; a clearer description of this process would improve the transparency of the blood bank operations.

18. The study acknowledges the importance of audits and updates to the MSBOS, but it would be valuable to include examples of how regular audits have directly impacted wastage reduction at KFMC.

19. The discussion of expired products could be more specific—are there particular types of blood products that are more prone to expiry, and if so, how can inventory strategies be adjusted to address this issue?

20. The conclusion section briefly mentions the need for more research, but it could be expanded to highlight the practical implications of this study for other hospitals with similar challenges.

Reviewer #2: Comment (01) The reference list must be improved to be up-to-date and to be close with the core of the topic

Comment (02) The citation must be as listed in the ref. in ascending order

Comment (03) The lack of section titled research gap make the appear the author efforts in weak

Comment (04) Where is the novelty of this work ? Motivation?

Comment (05) The abstract must be follow IMRaD manner

Comment (06) Where is the state-of-the-art for different teaching methods? To assign the proposed method in its place? Therefore, need your opinion in dividing this method to classes.

Comment (07) What is the robustness and the advantages of the suggested method over other existing methods?, especially if indicate the research gap

Comment (08) The resolution of pictures need enhancing to increase than 1280DPI.

Comment (09) Equations must be numbered and arranged in ascending order.

Comment (10) What is new in the paper? Motivation? Challenge?

Comment (11) The authors should discuss potential applications of the results obtained.

Comment (12) What is the new or state of the art

Comment (13) Can summarize the data in the conclusion section to be tabulated if allowable

Comment (14) Give the pseudocode for steps of using the proposed steps to meet the objective.

Comment (15) Minimum grammar required

Comment (16) Give the pseudocode for steps of using the proposed steps to meet the objective.

Comment (17) How did you manage the experimental biases and errors?

Comment (18) Did you consider numerical problems and errors?

Reviewer #3: The manuscript overall contributes valuable data on the performance of blood bank in a major tertiary hospital, a context where such operational studies remain scarce. However, there are some minor revision that would improve the manuscript. Here the following major comments:

1-While the study appropriately uses descriptive statistics and non-parametric tests, the analysis stops short of deeper interpretation. For example, Kruskal–Wallis tests show significant interdepartmental differences, but these findings are not clearly contextualized.

2-The discussion would benefit from quantitative comparison with regional and global benchmarks for each KPI. While the text mentions international studies, it lacks explicit numerical contrasts.

3-The terms wastage, discard, and expiry are used interchangeably in some sections, though the Methods define them differently. This inconsistency could confuse readers and affect reproducibility.

4-The tables are comprehensive but dense. The narrative repeats much of their content, which reduces readability. It would be better to replace repeated numeric details in the text with references to tables. Also, it is recommended to add one or two figures (e.g., a bar chart showing wastage rates per component)

Here some minor revision points:

1-Abstract: Suggest front-loading the quantitative findings (C/T, WAPI, ISI) earlier to emphasize outcomes. Consider shortening the methods description slightly.

2-Introduction: Lines 44–71 can be merged for smoother flow; these paragraphs overlap conceptually on wastage causes and KPI rationale.

2-Methods: Excellent clarity and ethics reporting. Ensure formulas are formatted uniformly (subscripts, fractions) according to PLOS ONE style guidelines.

3-Results: Some p-values are reported without corresponding effect or direction; provide concise interpretive statements where relevant.

4-Discussion: A stronger link between observed findings and management policy implications (e.g., training, audit frequency) would enhance impact.

**Do you want your identity to be public for this peer review?** For information about this choice, including consent withdrawal, please see our Privacy Policy

Reviewer #1: No

Reviewer #2: **Yes:** Ahmed M. Abed

Reviewer #3: No

---

## [Author Response · Author response to Decision Letter 1]

30 Dec 2025

Dear Editor,

Thank you for considering our manuscript #PONE-D-25-29668 for possible publication. Please find below our response to the reviewers’ comments ad verbatim.

Reviewer 1:

1. The introduction could be more concise, especially the sections that describe previous studies, to avoid redundancy with later discussions in the manuscript.

1. The introduction has been revised for greater conciseness and clarity, while retaining all essential context.

2. The background does not explicitly define the threshold values for C/T, ISI, and WAPI, which are crucial for understanding the operational standards and benchmarks used in the analysis.

2. We thank the reviewer for this valuable comment. We added threshold recommendations from NHS and AABB, where applicable, and reported from previous studies. These additions have been integrated into both the Methods and Discussion sections to provide clearer context and justification for the indicators used. The manuscript now clearly state the C/T threshold of <2 (line 114), the recommended ISI of approximately five days (line 123), and the commonly cited WAPI range of 2–7% (line 118).

3. There is no discussion on the limitations of using retrospective data; a brief acknowledgment of this limitation would strengthen the methodology section.

3. The limitation related to the retrospective study design is described in the discussion (lines 334). We also clarify additional constraints and explained how these factors may influence the study’s findings.

4. The manuscript does not clarify why data from January 2022 to December 2023 was selected as the study period, given the changes in blood demand and hospital practices over time.

4. Data from January 2022 to December 2023 were selected because the digital archived system used before this period used a different system, preventing reliable retrieval of earlier records. The two-year window provides a sufficiently robust timeframe to capture utilization and wastage patterns. We mentioned this limitation in line 81 of the manuscript.

5. The study claims the C/T ratio is 1.5, but a brief explanation of how this compares to international standards or previous studies would help contextualize the result.

5. We appreciate the reviewer’s suggestion. A statement comparing our C/T ratio of 1.5 with international benchmarks and previously published studies has now been added to the Discussion section line 294.

6. The methods section briefly mentions the Kruskal–Wallis test for comparing departmental utilization, but does not justify why this non-parametric test was chosen over other options, such as ANOVA.

6. The Kruskal–Wallis test was selected instead of ANOVA because the utilization data were not normally distributed, with skewness and department-specific outliers that violated ANOVA’s normality assumption as stated in line 106 of the Methods section.

7. While the paper discusses the concept of WAPI, it would be helpful to compare this metric with benchmarks from similar hospitals to evaluate the true significance of the reported values.

7. We thank the reviewer for this comment. A comparison of our WAPI values with benchmarks reported in similar hospital settings has now been added to the Discussion section (line 276).

8. The significance of the 0.6% discard rate is not well contextualized; the manuscript could benefit from a comparison to other similar healthcare facilities or regions.

8. We appreciate the reviewer’s observation. Contextual information regarding the 0.6% discard rate, including comparison with rates reported in similar healthcare settings, has now been incorporated into the Discussion section to strengthen the interpretation of this finding.

9. The relationship between the blood bank’s operational efficiency and the specific departments (ICU, surgery, etc.) could be explored in more detail to provide deeper insights into the causes of inefficiencies.

9. We thank the reviewer for this constructive suggestion. We have now expanded the Discussion to explore how operational efficiency may differ across major clinical departments and how these variations may contribute to differences in blood utilization and wastage patterns (lines 312 and 327).

10. The discussion of over-ordering, particularly in high-demand departments like the ICU, would benefit from further elaboration on possible solutions, such as implementing predictive algorithms for blood demand forecasting.

10. We thank the reviewer for this valuable suggestion. Additional text has been added describing possible solutions, including the use of demand-forecasting models, which have been reported to improve blood utilization efficiency (line 315).

11. The paper would benefit from including more specific recommendations for improving the MSBOS, particularly in departments with high over-ordering rates.

11. We thank the reviewer for this helpful recommendation. We included more specific, evidence-based strategies for improving the MSBOS, with particular attention to departments demonstrating higher over-ordering rates (line 324).

12. The methodology mentions the collection of anonymized data, but no information is provided about how the authors ensured the quality and consistency of the data across different departments.

12. All data were extracted directly by the hospital’s IT department, which included verification of completeness and deletion of missing or inconsistent fields to ensure dataset reliability.

13. While the study mentions that certain blood components were returned in acceptable condition, it would be useful to discuss whether these returned products were reallocated or disposed of.

13. We thank the reviewer for this comment. In the revised manuscript, we clarify that although the number of returned blood units was reported, these units were not included in the utilization or disposal analyses, as their subsequent reallocation or disposal could not be reliably tracked through the IT inventory system.

14. A potential limitation is the focus on a single hospital; the authors should address the generalizability of their findings to other hospitals in Saudi Arabia or internationally.

14. We agree with the reviewer’s comment, and this limitation was already acknowledged in the discussion line 334.

15. The manuscript provides data on blood product wastage but does not discuss potential consequences of waste, such as increased costs or impacts on patient care.

15. We thank the reviewer for this important observation. The potential consequences of blood-product wastage, including its economic implications and the indirect effects are addressed in lines 328.

16. The paper mentions that “not enough storage space” contributed to some discard rates; this could be a key area for future research on optimizing storage management.

16. We thank the reviewer for highlighting this important point. We have now acknowledged the relevance of storage limitations in contributing to discard rates and have noted this as an area warranting further investigation. This addition has been incorporated into the Conclusion to emphasize the need for future research on optimizing storage capacity and management practices.

17. There is little information on the process followed for managing returned blood products; a clearer description of this process would improve the transparency of the blood bank operations.

17. We thank the reviewer for this helpful observation. Additional clarification on the process for managing returned blood products has now been incorporated into the discussion line 250.

18. The study acknowledges the importance of audits and updates to the MSBOS, but it would be valuable to include examples of how regular audits have directly impacted wastage reduction at KFMC.

18. We thank the reviewer for this valuable suggestion. We have now expanded the Discussion and highlighted how MSBOS audits have contributed to a reduction in avoidable wastage (line 267)

19. The discussion of expired products could be more specific—are there particular types of blood products that are more prone to expiry, and if so, how can inventory strategies be adjusted to address this issue?

19. We have expanded the Discussion to specify that platelets were the most blood products prone to expiry and outlined inventory strategies that may mitigate this issue.

20. The conclusion section briefly mentions the need for more research, but it could be expanded to highlight the practical implications of this study for other hospitals with similar challenges.

20. We thank the reviewer for this constructive suggestion. The Conclusion has been edited to clearly outline the practical implications of our findings and how they may inform operational improvements in other hospitals facing similar inventory and utilization challenges.

Reviewer 2:

Comment (01) The reference list must be improved to be up-to-date and to be close with the core of the topic

Answer (01): We thank the reviewer for this comment. The reference list has been carefully reviewed and updated where possible. The majority of cited studies are from the past 10 years, with particular emphasis on the most recent five years; exceptions were made for seminal publications that provide original definitions or foundational concepts, where citation of the original work was necessary.

Comment (02) The citation must be as listed in the ref. in ascending order

Answer (02): We thank the reviewer for this observation. References are ordered in ascending order; however, in some instances—particularly within the Discussion—previously cited studies from the Introduction were referenced again, resulting in non-sequential citation numbering in the manuscript.

Comment (03) The lack of section titled research gap make the appear the author efforts in weak

Answer (03): We appreciate the reviewer’s suggestion. However, according to PLOS ONE author guidelines, a dedicated heading titled “Research Gap” is not required. Nevertheless, the research gap has been explicitly stated in both the Abstract (lines 1 and 2) and with more details in the Introduction (lines 64-68), ensuring that the study rationale is clearly articulated.

Comment (04) Where is the novelty of this work ? Motivation?

Answer (04): The novelty of this study lies in being the first study to jointly evaluate the ISI alongside WAPI and the C/T ratio as comprehensive indicators of blood bank performance within the Saudi healthcare context. This integrated assessment provides a more holistic evaluation of inventory efficiency than previously reported regional studies.

Comment (05) The abstract must be follow IMRaD manner

Answer (05): The abstract have been revised to follow the IMRaD format.

Comment (06) Where is the state-of-the-art for different teaching methods? To assign the proposed method in its place? Therefore, need your opinion in dividing this method to classes.

Answer (06): We thank the reviewer for this comment. Our study is a retrospective observational analysis using descriptive statistical methods and does not propose or evaluate a teaching or instructional method. Therefore, concepts such as state-of-the-art or classification of methods are not directly applicable to the scope of this work. We have clarified the study design and objectives to emphasize that the aim is to analyze and benchmark blood bank performance indicators.

Comment (07) What is the robustness and the advantages of the suggested method over other existing methods?, especially if indicate the research gap

Answer (07): We thank the reviewer for this comment. As this is a retrospective observational study, the use of descriptive statistics represents the standard and most appropriate approach for calculating established wastage and utilization indices. This allows direct comparability with published data from other institutions and addresses the identified local evidence gap.

Comment (08) The resolution of pictures need enhancing to increase than 1280DPI.

Answer (08): The original manuscript contained only tables and no figures. However, in response to a comment from another reviewer, two charts have now been added, and we ensured that they are presented in high resolution and suitable quality for publication.

Comment (09) Equations must be numbered and arranged in ascending order.

Answer (09): All equations have now been properly numbered and formatted in accordance with the PLOSONE style guidelines.

Comment (10) What is new in the paper? Motivation? Challenge?

Answer (10) Our findings demonstrate that the blood bank inventory management system at KFMC operates with a high degree of efficiency, achieving effective utilization of approximately 95% of blood components with minimal wastage. Additionally, only 3.8% of issued units were returned in good condition, reflecting optimized operational practices. Collectively, these results establish a local evidence base and lay the groundwork for future benchmarking and optimization strategies. Challenges are also illustrated in the discussion.

Comment (11) The authors should discuss potential applications of the results obtained.

Answer (11): The current study serves as a valuable benchmark for future research on blood bank inventory systems by providing local benchmarks not available before.

Comment (12) What is the new or state of the art

Answer (12): We thank the reviewer for this comment. This study is a retrospective observational analysis using descriptive statistics and does not include state-of-the-art technology.

Comment (13) Can summarize the data in the conclusion section to be tabulated if allowable.

Answer (13): The data have been summarized in Figures 4 and 5, and the Conclusion section has been revised accordingly. Adding a table to the Conclusion was not pursued, as this does not align with PLOS ONE formatting requirements.

Comment (14) Give the pseudocode for steps of using the proposed steps to meet the objective.

Answer (14): We thank the reviewer for this comment. This work is a retrospective descriptive study and does not involve algorithm development, programming, or Python-based analyses; therefore, providing pseudocode is not applicable. The steps undertaken to meet the study objectives are instead described narratively in the Methods section, in line with standard reporting practices for observational-retrospective research.

Comment (15) Minimum grammar required

Answer (15): The manuscript has been reviewed for grammatical inaccuracies.

Comment (16) Give the pseudocode for steps of using the proposed steps to meet the objective.

Answer (16): We thank the reviewer for this comment. This work is a retrospective descriptive study and does not involve algorithm development, programming, or Python-based analyses; therefore, providing pseudocode is not applicable. The steps undertaken to meet the study objectives are instead described narratively in the Methods section, in line with standard reporting practices for observational-retrospective research.

Comment (17) How did you manage the experimental biases and errors?

Answer (17): We thank the reviewer for this query. Anonymized data were extracted by the hospital IT department prior to analysis. As the data did not meet normality assumptions, the Kruskal–Wallis test was appropriately applied to compare outcomes across departments, followed by Dunn’s post hoc test with Bonferroni correction to identify specific interdepartmental differences. A two-sided p-value of <0.05 was considered statistically significant. These details are described in the methods section.

Comment (18) Did you consider numerical problems and errors?

Answer (18): Standard data-cleaning procedures were applied prior to analysis, including checks for missing values, duplicate records, and implausible entries. All statistical analyses were conducted using validated software to minimize numerical errors.

Reviewer 3:

The manuscript overall contributes valuable data on the performance of blood bank in a major tertiary hospital, a context where such operational studies remain scarce. However, there are some minor revision that would improve the manuscript. Here the follow

---

## [Decision Letter · Decision Letter 1]

23 Feb 2026

Article Title: Reducing Blood Product Wastage: Insights from a Retrospective Study in a Tertiary Health Facility

PONE-D-25-29668R1

Dear Dr. Alsughayyir,

We’re pleased to inform you that your manuscript has been judged scientifically suitable for publication and will be formally accepted for publication once it meets all outstanding technical requirements.

Kind regards,

Ramya Iyadurai

Academic Editor

PLOS One

Additional Editor Comments (optional):

none

Reviewers' comments:

Reviewer's Responses to Questions

**Comments to the Author**

Reviewer #2: All comments have been addressed

Reviewer #3: All comments have been addressed

2. Is the manuscript technically sound, and do the data support the conclusions?

Reviewer #2: Partly

Reviewer #3: Yes

3. Has the statistical analysis been performed appropriately and rigorously?

Reviewer #2: Yes

Reviewer #3: Yes

4. Have the authors made all data underlying the findings in their manuscript fully available?

Reviewer #2: Yes

Reviewer #3: Yes

5. Is the manuscript presented in an intelligible fashion and written in standard English?

Reviewer #2: No

Reviewer #3: Yes

Reviewer #2: (No Response)

Reviewer #3: (No Response)

**Do you want your identity to be public for this peer review?** For information about this choice, including consent withdrawal, please see our Privacy Policy

Reviewer #2: No

Reviewer #3: **Yes:** Dr. Wafaa Alhazmi
